# Calibration and Validation of a Cone Crusher Model with Industrial Data

**Robson A. Duarte** [1,2] , **André S. Yamashita** [3,*] , **Moisés T. da Silva** [4] , **Luciano P. Cota** [4]
**and Thiago A. M. Euzébio** [4]

1    Programa de Pós-Graduação em Instrumentação, Controle e Automação de Processos de Mineração,
     Universidade Federal de Ouro Preto e Instituto Tecnológico Vale, Ouro Preto 35400-000, Brazil
2    Vale S.A., Canaã dos Carajás 68537-000, Brazil; robson.duarte1@vale.com (R.A.D)
3    Department of Computer Science, Electrical and Space Engineering, Control Engineering Group,
     Luleå University of Technology, 97187 Luleå, Sweden
4    Instituto Tecnológico Vale (ITV), Ouro Preto 35400-000, Brazil; moises.silva@pq.itv.org (M.T.d.S.);
     luciano.p.cota@itv.org (L.P.C.); thiago.euzebio@itv.org (T.A.M.E.)
*    Correspondence: andre.yamashita@ltu.se

**Abstract:** This paper reports the calibration and validation of a cone crusher model using industrial data. Usually, there are three calibration parameters in the condensed breakage function; by contrast, in this work, every entry of the lower triangular breakage function matrix is considered a calibration parameter. The calibration problem is cast as an optimization problem based on the least squares method. The results show that the calibrated model is able to fit the validation datasets closely, as seen from the low values of the objective function. Another significant advantage of the proposed approach is that the model can be calibrated on data that are usually available from industrial operation; no additional laboratory tests are required. Calibration and validation tests on datasets collected from two different mines show that the calibrated model is a strong candidate for use in various dynamic simulation applications, such as control system design, equipment sizing, operator training, and optimization of crushing circuits.

**Keywords:** digital twin; cone crusher; model calibration; optimization; calibration based on industrial data

## 1. Introduction

Process modeling and simulation have considerable importance in the mineral processing industry. Accurate dynamic models of a mineral processing plant are helpful in control system development, equipment sizing, and operator training [1]. Currently, with the introduction of the concept of Industry 4.0 (a term used to describe the widespread integration of information and communication technology in industry [2]), process simulation is even more relevant because it can work as a process digital twin to find the optimal operating conditions [3], train reinforcement learning agents [4], or predict faults in equipment [5]. Among the different pieces of equipment that comprise a mineral processing plant, the cone crusher plays a prominent role in the comminution of run-of-mine ore.

Several approaches for modeling cone crushers are available in the literature. Whiten [6] used mass balance as well as classification and breakage matrices to model a crushing circuit. Whiten's model was extended by Lynch [7] to include electric current prediction for the crusher. Additional particle breakage equations were introduced by Whiten et al. [8] to offer alternative models for validating crushing data from different mines. Andersen [9] extended Whiten's classification function to include crusher liner characteristics and used $t_{10}$, the percentage of product passing through a sieve with a size equal to one-tenth of the original particle size, to predict breakage. In the model presented by Evertsson [10], the compression ratio, a parameter calculated based on cone crusher geometry data, helps to determine the breakage and classification matrices.

In the works cited above, the cone crusher is treated as a single breakage section and a classifier. Herbst and Oblad [11] proposed a different approach, in which the crushing chamber is divided into several zones. In the same vein, Atta et al. [12] used different breakage and selection parameters in each zone and, later on, these authors also introduced current and power consumption models [13]. Even more detailed modeling attempts consist of using models based on the discrete element method. Taking advantage of increases in computational power, this numerical technique is now also used to model the dynamic behavior of cone crushers [14,15].

From a practical perspective, the cone crusher model proposed by Whiten [6] provides a reasonable compromise between representability and simplicity [16]. More complex models may require information about the crusher geometry, and the subsequent calculation is time consuming [17]. Research on advanced control development [18], artificial intelligence applications [3], and optimization scheme comparison [19] has relied on Whiten's model. Moreover, the cone crusher blocks in many commercial simulators (e.g., IDEAS®—Andritz Automation, Mimic®—Emerson, and DynSim®—Aveva) are based on Whiten's model. Such dynamic simulation software is in high demand in digital twin applications, control system design, operator training, and flowsheet evaluation.

Model selection is only the first step in reliably simulating the behavior of a process. The second step consists of calibrating the model for its specific purposes (modeling a given plant, operational scenario, or dataset). Usually, the parameters of the breakage function are calibrated using the results of laboratory impact breakage testing, which is a time-consuming task [20]. Some alternative calibration approaches include trial and error, statistical inference, and a suitable optimization strategy [21]. Among these alternatives, optimization stands out in that it yields near-optimal solutions within a feasible amount of time for practical applications and offers a structured and replicable framework [22,23].

The advantages of model calibration based on optimization techniques have been established in the literature [22,23]. In 1984, Klimpel and Austin stated their preference for a condensed breakage function model, arguing that solving the calibration problem with an uncondensed model demands too much computational power [24]. Over the years, however, advances in computational capabilities may have made this argument obsolete, and allowing more degrees of freedom in the calibration problem may result in more precise and accurate crushing models, yielding an average error and a standard deviation smaller than 0.5% and 3.0%, respectively, between the predicted and measured particle size distributions (PSDs), as obtained by Whiten [6]. To the best of our knowledge, the most recent works that report the calibration of Whiten's cone crusher model date from the 1990s. This situation indicates that either the scientific community is satisfied with the available calibration strategies for Whiten's cone crusher model or no effort has been made to improve the calibration accuracy. We believe that the latter is the case, and thus, we propose a new calibration strategy for the cone crusher model.

In this paper, the calibration problem for Whiten's cone crusher model is cast as an optimization problem. The sum of the squared errors between the measured and estimated cumulative PSD values is minimized using a sequential quadratic programming algorithm. Instead of the condensed parameter breakage matrix model proposed by Whiten [6], the uncondensed—or full-matrix—model, in which all elements of the breakage matrix are calibration parameters, is used. This adds more degrees of freedom to the calibration problem. Operating data from the S11D and Serra Leste iron mines, operated by Vale S.A. in Brazil, were collected for model calibration and validation. Only data that are regularly collected during operation (PSD, closed side setting, and throughput) are necessary for the calibration strategy; additional laboratory test data, which might be difficult to obtain [25,26], are unnecessary.

The remainder of this paper is structured as follows: Section 2 summarizes the state-of-the-art of model calibration techniques in the mineral processing literature. Whiten's cone crusher model is described in Section 3. The calibration problem and the proposed

solution are defined in Section 4. Section 5 presents the calibration and validation results obtained on industrial datasets. Finally, Section 6 concludes the paper.

## 2. Literature Review

This section summarizes optimization-based calibration studies from the mineral processing literature, specifically studies on the first principles or empirical models of crushing, grinding and flotation equipment calibrated using optimization strategies.

An autogenous/semi-autogenous grinding mill model was calibrated by Perez et al. [27]. The calibration problem was cast as an optimization problem in which a global search method was used to minimize the difference between the predicted and measured values of the PSD. The static responses were validated using industrial data and showed accurate values for key variables (e.g., size distributions, hydrocyclone pressure, circulating load, mill filling level, and mill power drawn). Esnault et al. [28] developed and calibrated a population balance model for particle size prediction in compression grinding. The results showed that on the one hand, the model required extensive datasets collected under different operating conditions for proper calibration, but on the other hand, once calibrated, the model offered accuracy sufficient for applications related to the design optimization of mills with complex geometry, such as vertical mills and Horomills. Klimpel and Austin [24] proposed the calibration of specific rates of breakage and breakage distribution functions in grinding circuits using condensed parameters. According to the authors, calibrating all parameters of the equations is computationally expensive and does not result in physically interpretable parameters, which might indicate that the calibration problem is underdetermined.

High-pressure grinding rolls (HPGRs) are equipment used in some grinding circuits. Torres and Casali [29] estimated the parameters of an HPGR model using the gradient descent algorithm, and Hasanzadeh and Farzanegan [30] used a genetic algorithm. Daniel and Morrell [31] studied the modeling, calibration and scale-up of the HPGR process. In the calibration step, laboratory-scale and pilot plant data were collected and utilized to obtain the parameters of models of the PSD, throughput, and power drawn. The calibration problem was cast as a least squares optimization problem. The scaling-up procedure was performed based on a particular mechanical property of the HPGR (the ratio of the roll gap to the diameter), which was assumed to remain constant at any operating scale. Similarly, Campos et al. [32] also cast the calibration problem as a least squares optimization problem. The objective function was designed to consider the logarithms of the measured and estimated values of the cumulative PSD, and the resulting problem was solved using a direct search algorithm from *fminsearch* in MathWorks MATLAB®. Anticoi et al. [26] analyzed the influence of the type of material on the parameters of the breakage function in a milling circuit. The parameters were calibrated against data from piston press tests and single compression strength tests; the optimal parameters were obtained by minimizing the root mean square error between the measured and calculated values. Ambiguous results led the authors to conclude that numerical algorithms may not be the best option for calibration in the presented scenario. In a fashion similar to their previous study, Anticoi et al. [33] also studied the influence of the operating conditions of a pilot-scale HPGR on the breakage function parameters.

Calibrating flotation process models has proven to be a complex task. Seppälä et al. [23] developed and calibrated a dynamic model of a flotation circuit based on pilot-plant-scale data. Model calibration was performed by trial and error. The reported results indicate that the model allowed a better understanding of the effects of different feed materials and pH levels on the mineral concentration in the product. However, the authors noted that the calibration process should be performed by employing optimization routines to increase model adaptability to different plants and to guarantee that the calculated parameters are optimal. Yianatos et al. [25] developed a model for flotation processes based on industrial data from large flotation cells. The flotation kinetics is usually affected by both mineral type and operating conditions. In Yianatos' model, the effects of the two sources are treated

independently. Model calibration requires an extensive collection of data from laboratory tests and plant operation. Karr and Yeager [22] explored the application of a genetic algorithm for the calibration of a flotation column model. The authors tailored the fitness function of the algorithm to suit the calibration problem at hand, showcasing the flexibility of the technique. It was concluded that a model calibration method based on heuristic optimization is faster than, and as accurate as, statistical and trial-and-error-based methods.

Notably, reported calibration attempts for cone crusher models are remarkably scarce. Whiten [6] proposed solving the calibration problem for a cone crusher and vibrating screen model using nonlinear least squares methods. Industrial data from Mount Isa Mines (Australia) and North Broken Hill Consolidated (Australia) were used for calibration. The author reported an average error of 0.5% with a standard deviation of 3.0% between the predicted and measured PSDs. Machado Leite [34] solved the model calibration task for a cone crusher and vibrating screen process by fitting the predicted data to cone crusher data from the literature. The author did not disclose other details about the calibration method. King [35] calibrated Whiten's model [6] as implemented in commercial modular simulation software. The initial guesses for the parameters were based on guidelines from the literature (cone crusher type, standard vs. short-head), and the best fit was determined by trial and error. The calculated PSD values were compared to operating data from Rössing Uranium Ltd. (Namibia), and no quantitative information was given; however, the authors claimed that the fit was good. The calibrated crusher model, together with a calibrated screen model, was used in the commercial software to simulate the secondary, tertiary and quaternary crushing circuits from Rössing Uranium Ltd. More recent references report the calibration of the population balance model [36] and the bonded particle model [14] using optimization techniques. Such models are competing approaches for the discrete-element-method-based modeling of cone crushers. Due to their unreasonable computational demand, which make them infeasible for use in process control applications, discrete-element-method-based models are beyond the scope of this paper.

## 3. Description of the Cone Crusher Model

In the mineral processing industry, it is usually necessary to reduce the particle size of raw ores in order to extract the minerals and metals of interest. The comminution circuit of a mineral processing plant, comprised of crushing and grinding stages, is responsible for the mechanical size reduction of the raw ore. The cone crusher, the main equipment of a crushing circuit, can accomplish size reductions in the range of 250 mm to 10 mm. The working principle of a cone crusher is that a cone-shaped mantle moves eccentrically and, when the mantle moves away from the concave during its spin cycle, rock particles fall from the feeding area by gravity; as the distance between the mantle and the concave decreases, the particles are compressed and crushed [37]. Figure 1 shows a schematic representation of a cone crusher, in which $b$ indicates the bed thickness and $s$ indicates the eccentric throw or stroke, defined by the difference between the maximum and minimum distances between mantle and conclave, indicated by *OSS* and *CSS*, respectively. Therefore, $s = OSS - CSS$.

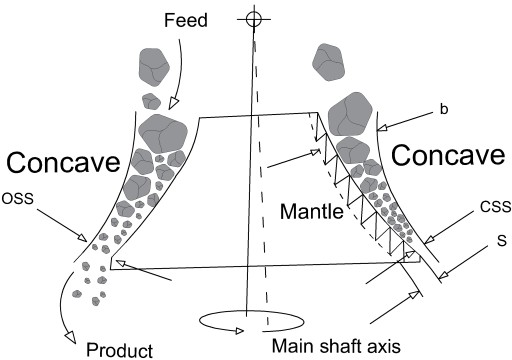

**Figure 1.** Schematic representation of a cone crusher, extracted from Yamashita et al. [37].

Figure 2 illustrates a block diagram representation of Whiten's cone crusher model. The feed stream is denoted by **f**; the material stream inside the cone crusher, which consists of the feed stream added to the pre-existing circulating material, is denoted by **x**; and the breakage and classification functions are represented by **B** and **C**, respectively. At **C**, material with a particle size below the classification values leaves the crusher as the product (**p**), and the remainder of the material goes to **B**. The number of classes ($n_f$) into which the material stream is divided defines the dimensionality of **x**, **f**, and **p** (**x**, **f**, **p** $\in \mathbb{R}^{n_f}$). This value consists of the number of particle size ranges used to define the PSD of the stream. In theory, the ratio between one class and the next should be $\sqrt{2}$, which means that the sieve size difference and, consequently, the particle size difference between adjacent classes should also be $\sqrt{2}$. However, in practice, the number of classes and their values are defined by the available sieve sizes [20].

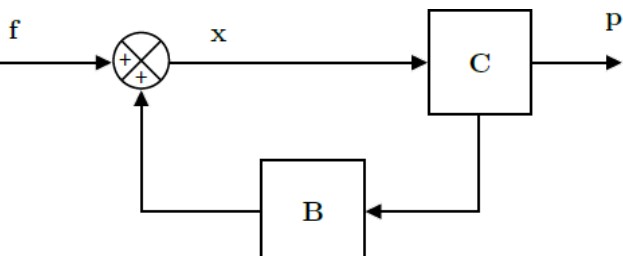

**Figure 2.** Block diagram representation of the cone crusher model.

The datasets studied in this work contain measurements of the nominal values of the sieve aperture diameters ($D_i$) and the cumulative passing PSD values of the feed ($F_i$), where $i$ denotes the size class, $i = 1, \ldots, n_f$. Therefore, to comply with Whiten's model, these values are converted into the actual particle size values ($d_{pi}$) and the incremental PSD values of the feed ($f_i$) in the model calibration step using Equations (1) and (2). In (1), for $i = n_f$, we have $D_{i+1} = \frac{D_i}{\sqrt{2}}$. Note that this step is unnecessary if $f_i$ and $d_{pi}$ are measured directly.

$$d_{pi} = \sqrt{D_i \times D_{i+1}}, i < n_f. \tag{1}$$

$$f_i = \begin{cases} F_i - F_{i+1}, & i < n_f \\ F_i, & i = n_f \end{cases}. \tag{2}$$

The steady-state of Whiten's cone crusher model results from the mass balance per material class size applied to a cone crusher [16]:

$$\frac{d\mathbf{m}(t)}{dt} = \mathbf{f}(t) - \mathbf{p}(t) - \omega \times \zeta(t)(\mathbf{S} - \mathbf{BS})\mathbf{m}(t), \tag{3}$$

where $\zeta(t)$ represents the ore hardness, **S** is called the selection function and represents the specific breakage rate, **B** is called the breakage function, $\omega$ if the operational frequency of the cone crusher, and **m** represents the mass vector in the cone crusher. The product **p** is considered to be proportional to the mass components, that is, $\mathbf{p} = \mathbf{Q}\,\omega \times \mathbf{m}$, where **Q** is a diagonal matrix that represents each element in the specific discharge rate. The unit of the elements of vectors **f** and **p** is given by $t/h$, $\omega$ is measured in $1/h$ and, the variable $\zeta(t)$ and the elements of matrices **S**, **B** and **C** are dimensionless.

The product PSD is determined by the feed size distribution and the classification and breakage functions. The steady-state solution to (3) is obtained by setting the left-hand side to zero and expressing **p** as a function of **f** [16]. After some algebraic manipulation, according to Whiten [6], the incremental PSD of the product stream (**p**) is given by:

$$\mathbf{p} = (\mathbf{I} - \mathbf{C})\,(\mathbf{I} - \mathbf{B}\,\mathbf{C})^{-1}\,\mathbf{f}, \tag{4}$$

where **I** is an identity matrix of appropriate dimensions, **C** is the classification matrix defined as $\mathbf{S}(\mathbf{S} + \mathbf{Q})^{-1}$, and **B** is the breakage matrix. The accumulated particle size $P_i$ is calculated as the sum of the $p_i$ from $i = 1$ up to the desired class:

$$P_i = \begin{cases} 100, & \text{for } i = 1 \\ P_{i-1} - p_{i-1}, & \text{for } i > 1 \end{cases}. \tag{5}$$

### 3.1. Classification Function

In Whiten's cone crusher model, the classification function represents the probability that material of a given particle class will either become product or undergo breakage. Figure 3 shows how the classification function is determined in terms of parameters $K_1$, $K_2$, and $K_3$, and (6) shows the algebraic representation of the classification function, in which $c_{ii}$ is the $i$-th element of a diagonal matrix **C** ($\mathbf{C} \in \mathbb{R}^{n_f \times n_f}$) [38]. If $d_{p_i}$ is lower than or equal to $K_1$, then particles in this class are not crushed any further and leave the cone crusher as product. If $d_{p_i}$ is greater than or equal to $K_2$, then the particle class undergoes additional crushing. $K_3$ describes the shape of the classification function, defining the probability of breakage for particles of intermediate classes.

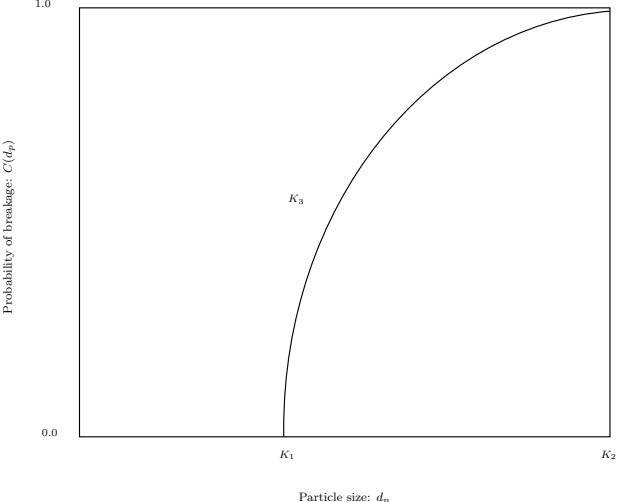

**Figure 3.** Schematic representation of the classification function.

$$c_{ii} = \begin{cases} 0, & \text{for } d_{p_i} \leq K_1 \\ 1 - \left( \dfrac{K_2 - d_{p_i}}{K_2 - K_1} \right)^{K_3}, & \text{for } K_1 < d_{p_i} < K_2 \\ 1, & \text{for } d_{p_i} \geq K_2 \end{cases}. \tag{6}$$

According to Napier-Munn et al. [38], the linear relationships between the parameters $K_1$ to $K_3$ and the cone crusher operating parameters are given by:

$$K_1 = \alpha_0 + \alpha_1\, CSS - \alpha_2\, TPH + \alpha_3\, F80 + \alpha_4\, LLEN, \tag{7}$$
$$K_2 = \beta_0 + \beta_1\, CSS + \beta_2\, TPH + \beta_3\, F80 - \beta_4\, LHR + \beta_5\, ET, \tag{8}$$
$$K_3 = \gamma_0, \tag{9}$$

where $CSS$ is the closed side setting (mm); $TPH$ is the throughput (dry t/h); $F80$ is the 80% passing size of the feed (mm); $LLEN$ is the length of the face of the mantle liner (mm); $LHR$ is the liner age (hours); $ET$ is the eccentric throw (mm); and $\alpha_0$ to $\alpha_4$, $\beta_0$ to $\beta_5$, and $\gamma_0$ are the calibration parameters of the classification function model. The signs of the parameters $\alpha_2$ and $\beta_4$ show the expected trends with the dominant variables of crusher set and throughput.

According to King [20], parameter $K_1$ ranges from about 0.5 to 0.95 and $K_2$ ranges from about 1.7 to 3.5. Usually, $K_3$ is approximately equal to 2, but can vary from 1 to 3.

### 3.2. Breakage Function

In Whiten's cone crusher model, the breakage function represents the probability that particles of a given class in a material stream will be broken into a smaller size. Equation (10) defines the lower triangular breakage matrix **B** ($\mathbf{B} \in \mathbb{R}^{n_f \times n_f}$) and represents the fraction of particles smaller than class size $w$, resulting from the breakage of particles of class size $z$, which occurs on each feed cycle. The elements of **B** are calculated as shown in (11), in which $b_{ij}$ is the element in row $i$ and column $j$ of matrix **B** and gives the percentage of particles of size class $j$ that will belong to particle class $i$ after breakage and $B(d_{p_i}; d_{p_j})$ is calculated as shown in (10). The sum of the elements in each column of **B** must be equal to 1.

$$B(w;z) = \Phi\left(\frac{w}{z}\right)^{\delta} + (1 - \Phi)\left(\frac{w}{z}\right)^{\sigma}. \tag{10}$$

$$b_{ij} = \begin{cases} B(d_{p_i}; d_{p_j}) - B(d_{p_{(i+1)}}; d_{p_j}), & \text{for } i \neq j \\ 1 - B(d_{p_{(i+1)}}; d_{p_j}), & \text{for } i = j \end{cases}. \tag{11}$$

$$\sum_{j=1}^{n_x} b_{ij} = 1, \forall i, 1 \leq i \leq n_x. \tag{12}$$

In (10), $\Phi$ denotes the fraction of child particles in the finer fraction. $\delta$ and $\sigma$ describe the larger particles produced by tensile stress and the smaller products produced by intense compressive stress, respectively [20]. The parameters $\Phi$, $\delta$, and $\sigma$ depend on the particular application and should be calibrated. According to King [20], for standard crushers, the $\delta$ and $\sigma$ values must be equal to 0.5 and 4.5, respectively.

Equations (10)–(12) express the breakage matrix in a condensed parameter form, as only three parameters define all elements of the matrix. Klimpel and Austin [24] used this condensed breakage function to reduce the number of calibration parameters. These authors argued that the calibration problem (cast as an optimization problem) can be solved more efficiently with fewer calibration parameters. Similarly, Machado Leite [34] argued that the condensed form of the breakage function is more tractable than the uncondensed form and that the calibration parameters have greater physical meaning. Nevertheless, in this paper, we choose to increase the number of degrees of freedom in the breakage function at the expense of additional complexity. Instead of calibrating only the parameters $\Phi$, $\delta$, and $\sigma$, we propose to calibrate every element of the lower triangular breakage matrix. The size of the lower triangular breakage matrix is $n_f \times n_f$, resulting in $n_f(n_f + 1)/2$ calibration parameters. Note that, in this case, the constraint defined in (12) still holds.

We demonstrate that, by taking advantage of current computational capabilities, calibrating every element of the breakage matrix, as opposed to what was done by Klimpel and Austin [24] and Machado Leite [34], is not only computationally feasible but also results in a more precise and accurate cone crusher model. Additionally, cone crusher models based on the lower triangular breakage matrix, despite lacking physically meaningful parameters, are valid alternatives for circuit design and process control applications [34].

## 4. Proposed Calibration Strategy

This section presents the calibration strategies for Whiten's cone crusher model formulated as a nonlinear programming model. For completeness, two variants of the calibration problem are defined, namely, (*i*) the problem with the full breakage matrix calibration strategy (FBS) and (*ii*) the problem with the condensed breakage matrix calibration strategy (CBS). The former is the strategy proposed in this work, whereas the latter is an established strategy in the literature [24,34]. For both calibration strategies (CBS and FBS), the resulting optimization problems are solved using a sequential quadratic programming (SQP)

method [39], which exhibits excellent performance for solving nonlinear optimization problems [40].

The cone crusher model presented in the previous section is a function of several parameters that must be calibrated. We ultimately wish to have a model that can predict the product PSD of a cone crusher, thus enabling the implementation of process control algorithms that rely on a plant model.

Hereafter, we use the following notation:

- $DS = \{1, \ldots, n_d\}$: set of $n_d$ datasets;
- $CL = \{1, \ldots, n_f\}$: set of $n_f$ classes used to describe the PSD of the material;
- $CSS_l$: closed side setting (CSS; mm) of the cone crusher for dataset $l$, $l = 1, \ldots, n_d$;
- $TPH_l$: throughput (dry t/h) for dataset $l$, $l = 1, \ldots, n_d$;
- $F80_l$: 80% passing size (mm) of the feed for dataset $l$, $l = 1, \ldots, n_d$;
- $LLEN_l$: length of the face of the mantle liner (mm) for dataset $l$, $l = 1, \ldots, n_d$;
- $LHR_l$: liner age (hours) for dataset $l$, $l = 1, \ldots, n_d$;
- $ET_l$: eccentric throw (mm) for dataset $l$, $l = 1, \ldots, n_d$;
- $\hat{P}_{i,l}(\theta_1)$, $P_{i,l}$: predicted and measured product PSDs, respectively, for $i = 1, \ldots, n_f$ and $l = 1, \ldots, n_d$;
- $\theta$: vector of decision variables;
- $LB$: lower bound on the decision variables;
- $UB$: upper bound on the decision variables.

The FBS optimization problem is formulated as follows:

$$\min_{\theta_1} \quad \sum_{l \in DS} \sum_{i \in CL} (\hat{P}_{i,l}(\theta_1) - P_{i,l})^2 \tag{13}$$

subject to:

$$\sum_{j \in CL} b_{ij} = 1, \qquad\qquad \forall i \in CL, \tag{14}$$

$$0.5\,CSS_l \leq (\alpha_0 + \alpha_1\,CSS_l - \alpha_2\,TPH_l + \alpha_3\,F80_l + \alpha_4\,LLEN_l) \leq 0.95\,CSS_l, \qquad \forall l \in DS, \tag{15}$$

$$1.7\,CSS_l \leq (\beta_0 + \beta_1\,CSS_l + \beta_2\,TPH_l + \beta_3\,F80_l - \beta_4\,LHR_l + \beta_5\,ET_l) \leq 3.5\,CSS_l \qquad \forall l \in DS, \tag{16}$$

$$LB \leq \theta_1 \leq UB. \tag{17}$$

The decision variables for the FBS problem are as follows: $\alpha_0, \ldots, \alpha_4$, used in the calculation of the parameter $K_1$ for the classification function; $\beta_0, \ldots, \beta_5$, used in the calculation of the parameter $K_2$ for the classification function; $\gamma_0$, used in the calculation of the parameter $K_3$ for the classification function; and $b_{ij}$, which represents the percentage of particles that are broken inside the cone crusher and change from particle class $j \in CL$ to particle class $i \in CL$. The resulting decision variable vector is $\theta_1 = \{\alpha_0, \ldots, \alpha_4 \cup \beta_0, \ldots, \beta_5 \cup \gamma_0 \cup b_{ij}\}$. The objective, as expressed in (13), is to minimize the sum of the squared errors (SSE) between the measured PSD ($P$) and the PSD calculated using the cone crusher model ($\hat{P}$), given by (4)–(12) for the FBS. For the CBS, the cone crusher model ($\hat{P}$) is given by (4)–(9) and (11)–(12). The constraints in (14) ensure that the sum of the entries in each column of **B** is equal to one. Constraints (15) and (16) define bounds on $K_1$ and $K_2$ based on empirical recommendations [20]. Finally, constraint (17) defines bounds on $\theta_1$.

The CBS optimization problem corresponds to a calibration strategy established in the literature [24,34]. In the CBS, the breakage function parameters appearing in (10) are calibrated. Thus, the decision variables $b_{ij}$ are removed, and three new decision variables are added: $\Phi$, $\delta$, and $\sigma$. The objective function is designed to minimize the SSE of a new $\theta$, represented by $\theta_2 = \{\alpha_0, \ldots, \alpha_4 \cup \beta_0, \ldots, \beta_5 \cup \gamma_0 \cup \Phi \cup \delta \cup \sigma\}$. The constraints are similar to those in the FBS problem except that in constraint (17), $\theta_1$ changed to $\theta_2$.

## 5. Results and Discussion

This section presents applications of the proposed calibration strategy for Whiten's cone crusher model. The case studies consider seven datasets consisting of operating data

from Vale S.A., a Brazilian mining company. The cone crusher throughput and the PSDs of the feed and product flows were collected for three different CSSs and throughputs of the Serra Leste iron mine (datasets 1 to 3); the PSD of the product flow was also collected at a CSS of 28 mm for four different *F*80 values from the S11D iron mine (datasets 4 to 7). Table 1 summarizes the datasets used for the calibration of cone crusher models. For Serra Leste datasets, note that the parameter F80 is not considered for calibration purposes. The variations by time of F80 are negligible for this crushing stage.

**Table 1.** Summary of plant name, cone crusher type and operating parameters for each dataset.

| Iron Mine | Cone Crusher | Dataset | *CSS* (mm) | *TPH* (t/h) | *F*80 (mm) |
|-----------|--------------|---------|------------|-------------|------------|
| Serra Leste | Metso HP400 | 1 | 35 | 883 | 102.36 |
| | | 2 | 38 | 986 | 102.36 |
| | | 3 | 41 | 998 | 102.36 |
| S11D | Sandvik CH660 | 4 | 28 | 368 | 45.35 |
| | | 5 | 28 | 368 | 73.03 |
| | | 6 | 28 | 368 | 30.89 |
| | | 7 | 28 | 368 | 27.79 |

Due to the limited availability of data obtained during plant operation and the lack of information regarding operating conditions, some of the parameters of the classification model in (6) were disregarded during the calibration procedure, namely, variables *LLEN*, *LHR*, and *ET*, which are related to parameters $\alpha_4$, $\beta_4$, and $\beta_5$, respectively. The decision variable vectors were adjusted accordingly.

For all datasets, the number of classes in the material stream was determined in a laboratory test, resulting in $n_f = 22$. Since datasets 1 to 3 and datasets 4 to 7 were collected from different iron mines, they were treated separately in the calibration and validation tests.

The FBS and CBS problems were solved using a commercial SQP algorithm (*fmincon*, MATLAB) on a computer with a 1.6 GHz CPU and 8 GB of RAM running MATLAB version 2019b on the 64-bit Windows 10 operating system. The FBS and CBS problems have 262 and 12 decision variables, respectively. The significant difference in the number of variables originates from the elements $b_{ij}$ of **B**, which correspond to $n_f(n_f + 1)/2 = 253$ parameters. The maximum number of iterations and the maximum number of function evaluations of the SQP algorithm were set to $10^{10}$, and all other settings were set to their default values.

In all of the calibration/validation scenarios studied here, the initial guess for the FBS ($\theta_1'$) was defined based on the work of Napier-Munn et al. [38] as follows:

$$\theta_1' = \begin{bmatrix} 0 & 1 & 0 & 0 & 0 & 2.43 & 0 & 0 & 2.3 & \mathbb{1}_{1,\frac{n_x(n_x+1)}{2}} \times 0.01 \end{bmatrix},$$

where $\mathbb{1}_{1,\frac{n_f(n_f+1)}{2}}$ represents a vector of ones with a size equal to $n_f(n_f + 1)/2$. The initial guess for the CBS ($\theta_2'$), when applicable, was defined based on the work of Napier-Munn et al. [38] and King [20] as follows:

$$\theta_2' = \begin{bmatrix} 0 & 1 & 0 & 0 & 0 & 2.43 & 0 & 0 & 2.3 & 0.3 & 0.5 & 4.5 \end{bmatrix}.$$

The lower and upper bounds of the optimization problems are as follows:

$$\begin{bmatrix} 0 \\ 0 \\ 1 \\ 0 \\ 0 \\ 0 \end{bmatrix} \leq \begin{bmatrix} \alpha_a \\ \beta_b \\ \gamma_0 \\ \Phi \\ \delta \\ \sigma \end{bmatrix} \leq \begin{bmatrix} \infty \\ \infty \\ 3 \\ 1 \\ \infty \\ \infty \end{bmatrix}, \tag{18a}$$

$$\begin{bmatrix} 0 \\ 0 \\ 1 \\ 10^{-3} \end{bmatrix} \leq \begin{bmatrix} \alpha_a \\ \beta_b \\ \gamma_0 \\ b_{ij} \end{bmatrix} \leq \begin{bmatrix} \infty \\ \infty \\ 3 \\ 1 \end{bmatrix}, \tag{18b}$$

where $a,b = 0,\ldots,3$, $i = 1,\ldots,n_f$, and $j = 1,\ldots,n_f$.

### 5.1. Evaluation of the FBS and CBS

Figure 4 illustrates the difference between the predictions of the product PSD obtained using the breakage function presented in (10) (CBS) and the breakage matrix defined in (11) (FBS) for calibration using dataset 1. In this case, the SSE values show that the difference is significant: $SSE = 98.66$ for the model calibrated using the CBS, and $SSE = 6.04 \times 10^{-5}$ for the model calibrated using the FBS. In fact, the curves marked with 'x' symbols (measured data) and 'o' symbols (data predicted using the FBS) are virtually interchangeable. This finding demonstrates that increasing the number of degrees of freedom of the model yields a better fit.

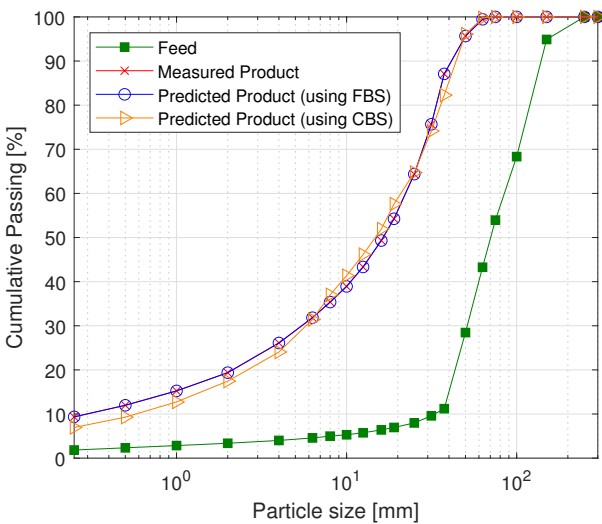

**Figure 4.** Measured and predicted PSD curves to compare the effect of the number of degrees of freedom of the breakage function on the model accuracy for dataset 1.

Table 2 shows that the FBS also yields better model fits than the CBS for calibration using the two other Serra Leste datasets and various combinations (investigated in Scenario 1 in the next subsection). The mean and standard deviation of the SSE for the FBS are 6.43 and 10.56, respectively, whereas the mean and standard deviation for the CBS are 267.90 and 250.22, respectively. The time required to solve the FBS optimization problem in the slowest case was approximately 3 min. Although this is a significant difference compared to the slowest solution time reported for the CBS (4 s), the magnitude of the computation time for the FBS is not relevant in practice for model applications. Moreover, calibration via the FBS is much faster than calibration by trial and error. More complete discussions of the SSE values in rows 1–3, rows 4–6, and row 7 of Table 2 are presented in Section 5.2.1 and Section 5.2.2, respectively. Table 3 shows the same comparison for S11D datasets. The mean and standard deviation of the SSE for the FBS are 124.34 and 236.82, respectively, whereas the mean and standard deviation for the CBS are 409.37 and 444.76, respectively. More complete discussions of the SSE values in rows 1–3, and row 4 of Table 3 are presented in Section 5.2.1 and Section 5.2.2, respectively.

**Table 2.** Summary of SSEs and computation times using the FBS and CBS (Serra Leste).

| Dataset(s) Used for Calibration | FBS | | CBS | |
|---|---|---|---|---|
| | SSE | Time [s] | SSE | Time [s] |
| 1 | $6.04 \times 10^{-5}$ | 20.72 | 98.66 | 1.43 |
| 2 | $3.01 \times 10^{-6}$ | 13.32 | 72.16 | 2.90 |
| 3 | $2.21 \times 10^{-6}$ | 12.92 | 76.10 | 1.53 |
| 1 and 2 | 22.51 | 180.39 | 407.95 | 1.94 |
| 1 and 3 | 1.26 | 79.04 | 308.75 | 1.51 |
| 2 and 3 | $1.28 \times 10^{-2}$ | 97.63 | 155.85 | 1.60 |
| 1, 2 and 3 | 21.23 | 58.07 | 755.82 | 3.76 |
| Mean (StDev) | 6.43 ($\pm$10.56) | 66.01 ($\pm$60.50) | 267.90 ($\pm$250.22) | 2.10 ($\pm$0.89) |

**Table 3.** Summary of SSEs and computation times using the FBS and CBS (S11D).

| Dataset(s) Used for Calibration | FBS | | CBS | |
|---|---|---|---|---|
| | SSE | Time [s] | SSE | Time [s] |
| 5 and 7 | 10.15 | 30.94 | 244.83 | 10.45 |
| 4 and 7 | 7.67 | 22.29 | 258.50 | 3.23 |
| 4 and 5 | $4.94 \times 10^{-2}$ | 56.52 | 70.15 | 2.99 |
| 4, 5, 6 and 7 | 479.51 | 42.11 | 1064.00 | 2.50 |
| Mean (StDev) | 124.34 ($\pm$236.82) | 37.96 ($\pm$14.79) | 409.37 ($\pm$444.76) | 4.79 ($\pm$3.78) |

*5.2. Calibration/Validation Using the FBS*

The analysis of the FBS, as a candidate strategy for developing cone crusher models based on industrial datasets, is divided into three scenarios. In Scenario 1, the model parameters are calibrated using datasets 1, 2 and 3 simultaneously and validated on each dataset one at a time. This scenario is simple in the sense that a model is both calibrated and validated using the same dataset. Ideally, this situation should be avoided; however, in industrial practice, additional datasets may be scarce. In Scenario 2, the model parameters are calibrated using two datasets (chosen from among either datasets 1–3 or datasets 4–7) and validated on a third dataset not used in the calibration step. Scenario 3, on the other hand, represents the usual practice of calibrating and validating a model on different datasets with an additional degree of complexity, as all datasets are obtained under different operating conditions. In Scenario 3, the model calibrated in Scenario 1 is used to predict PSD curves for different CSS values.

In all scenarios, for comparison, we plot the result of trial and error approach considering the recommended parameters values by King [20], i.e., $K_1 = [0.5 - 0.95]$, $K_2 = [1.7 - 3.5]$, $K_3 = [1 - 3]$, $\Phi = [0.1 - 0.9]$, $\delta = 0.5$ and $\sigma = 4.5$. In this case, the green region illustrate all available possibilities of result.

5.2.1. Scenario 1

In this scenario, the calibration procedure considers all available datasets from a single iron mine simultaneously. For the Serra Leste mine (datasets 1 to 3), Figure 5 shows that there is almost no difference between the predicted and measured PSD curves, even when the model is adjusted for datasets with different CSSs. The SSE values for validation using datasets 1, 2, and 3 individually are 13.40, 5.49, and 2.49, respectively. For the S11D mine (datasets 4 to 7), Figure 6 similarly shows that the predicted PSD is close to the measured PSD even though the $F80$ value is different for each dataset used. The SSE values for validation using datasets 4, 5, 6, and 7 individually are 60.24, 7.17, 273.80 and 138.29, respectively. Table 4 summarizes the calibrated parameters. Note that the variation of the calibrated parameters is due to the optimization algorithm's attempt to fit the data used in the calibration with the cone crusher model.

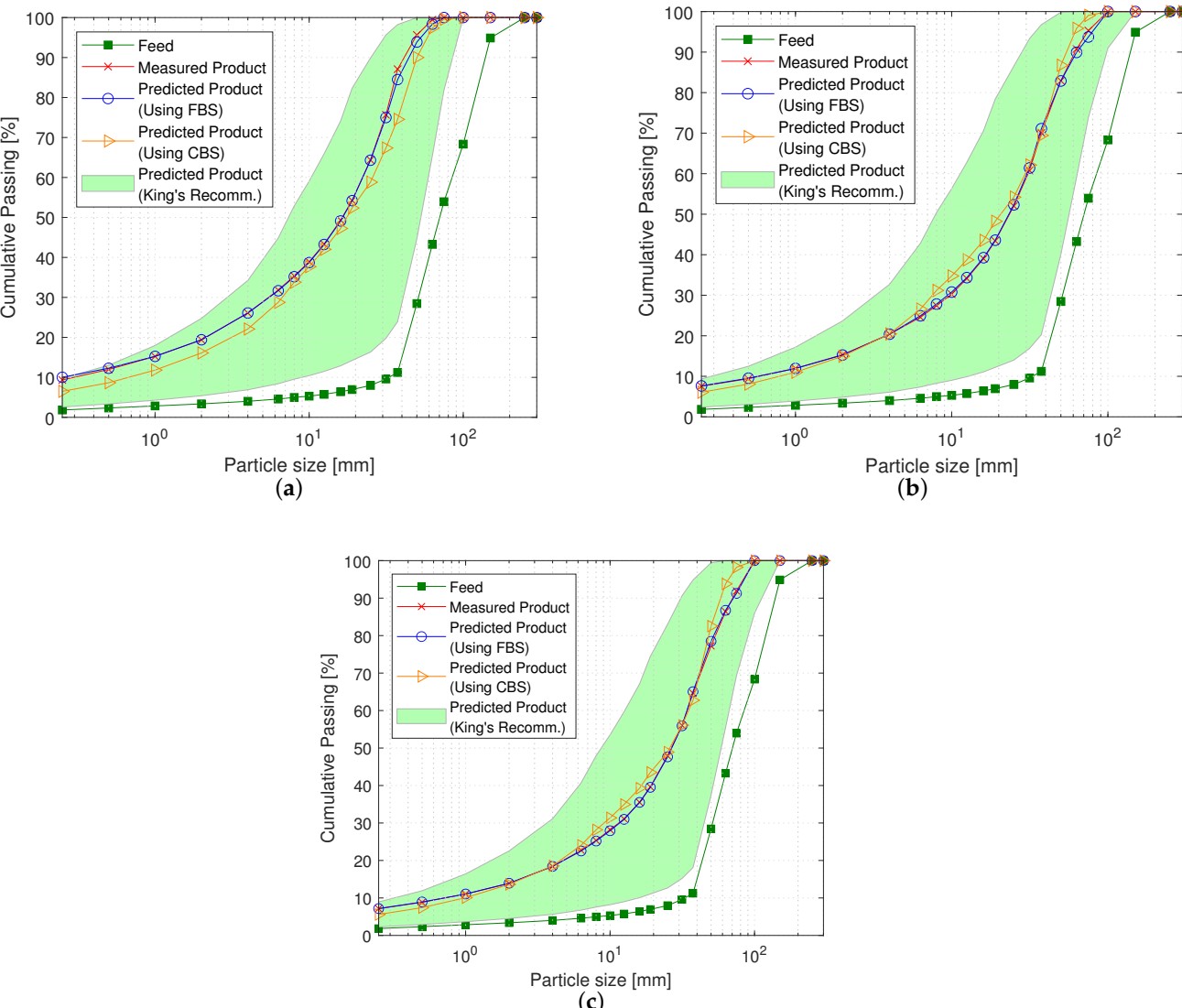

**Figure 5.** Measured and predicted PSD curves in Scenario 1 for the Serra Leste datasets. (**a**) PSD predicted using the model calibrated on datasets 1, 2 and 3. Validation with Dataset 1. (**b**) PSD predicted using the model calibrated on datasets 1, 2 and 3. Validation with Dataset 2. (**c**) PSD predicted using the model calibrated on datasets 1, 2 and 3. Validation with Dataset 3.

Using the CBS, the resulting calibrated parameters are those showed in Table 5. The SSE values for validation using datasets 1, 2 and 3 (Serra Leste) individually are 387.54, 170.70, and 197.59, respectively. The SSE values for validation using datasets 4, 5, 6 and 7 (S11D) individually are 146.29, 24.23, 579.91, and 313.60, respectively.

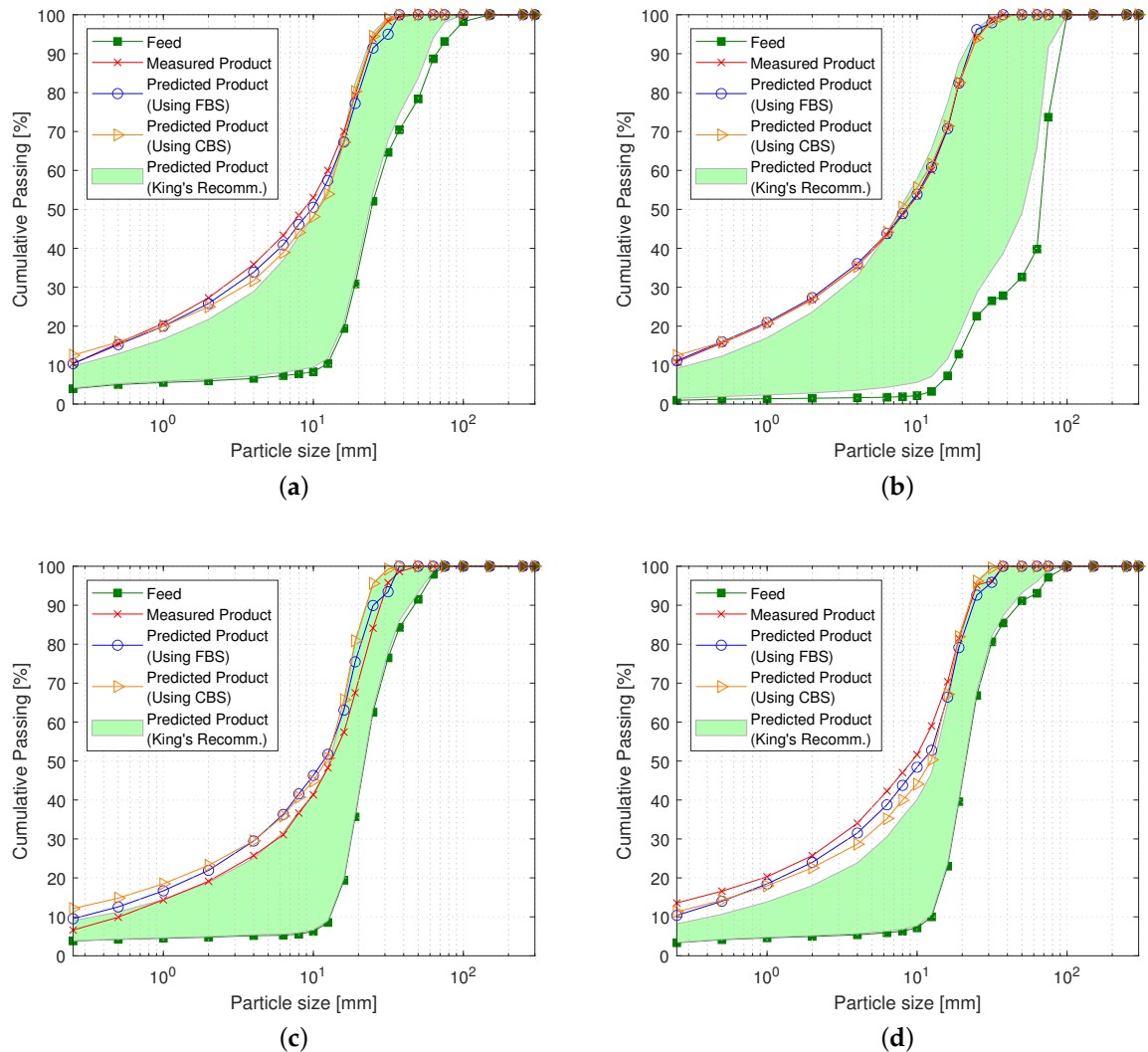

**Figure 6.** Measured and predicted PSD curves in Scenario 1 for the S11D datasets. (**a**) PSD predicted using the model calibrated on datasets 4 to 7. Dataset 4 is used for validation. (**b**) PSD predicted using the model calibrated on datasets 4 to 7. Dataset 5 is used for validation. (**c**) PSD predicted using the model calibrated on datasets 4 to 7. Dataset 6 is used for validation. (**d**) PSD predicted using the model calibrated on datasets 4 to 7. Dataset 7 is used for validation.

**Table 4.** Calibrated parameters in Scenario 1.

| Iron Mine | $\alpha_0$ | $\alpha_1$ | $\alpha_2$ | $\alpha_3$ | $\beta_0$ | $\beta_1$ | $\beta_2$ | $\beta_3$ | $\gamma_0$ |
|---|---|---|---|---|---|---|---|---|---|
| Serra Leste (1 to 3) | 0 | 0.905 | 0 | 0 | 0 | 0.096 | 0.095 | 0 | 3.000 |
| S11D (4 to 7) | 0.155 | 0.441 | 0 | 0.053 | 0.047 | 1.698 | 0 | 0 | 3.000 |

**Table 5.** Calibrated parameters in Scenario 1 (CBS).

| Iron Mine | $\alpha_0$ | $\alpha_1$ | $\alpha_2$ | $\alpha_3$ | $\beta_0$ | $\beta_1$ | $\beta_2$ | $\beta_3$ | $\gamma_0$ | $\Phi$ | $\delta$ | $\sigma$ |
|---|---|---|---|---|---|---|---|---|---|---|---|---|
| Serra Leste (1 to 3) | 0 | 1.375 | 0.018 | 0 | 0 | 3.060 | 0 | 0 | 3.000 | 1.000 | 0.506 | 211.441 |
| S11D (4 to 7) | 0.001 | 0.425 | 0 | 0.076 | 0.004 | 0.704 | 0.063 | 0.177 | 3.000 | 1.000 | 0.400 | 875.255 |

In this scenario, a compromise solution is expected since the cone crusher model is calibrated using datasets obtained under different operating conditions. Nevertheless, the curves in Figures 5 and 6 show that the PSDs from all sets are well adjusted by the model

for both iron mines. These findings demonstrate that the FBS-based cone crusher model will also behave well when multiple datasets from the same equipment but at different operating points are available for calibration. This is a common practical scenario when a model that is valid over a large operating range is desired. In the final scenario, we will evaluate whether the predicted cone crusher model can predict PSD curves for arbitrary CSS values. First, however, we consider the next scenario to evaluate how the model fares in calibration/validation trials using various combinations of the available datasets.

### 5.2.2. Scenario 2

In this scenario, a calibration and validation study was performed considering three datasets with different CSSs (for the data from Serra Leste) or different values of *F*80 (for the data from S11D). The use of combinations of 2 datasets for calibration and 1 for validation was investigated. Figures 7 and 8 illustrate the measured and predicted PSD curves for Serra Leste and S11D, respectively. The calibrated parameters obtained for the Serra Leste and S11D datasets are listed in Table 6. Again, note that the variation of the calibrated parameters is due to the optimization algorithm's attempt to fit the data used in the calibration with the cone crusher model.

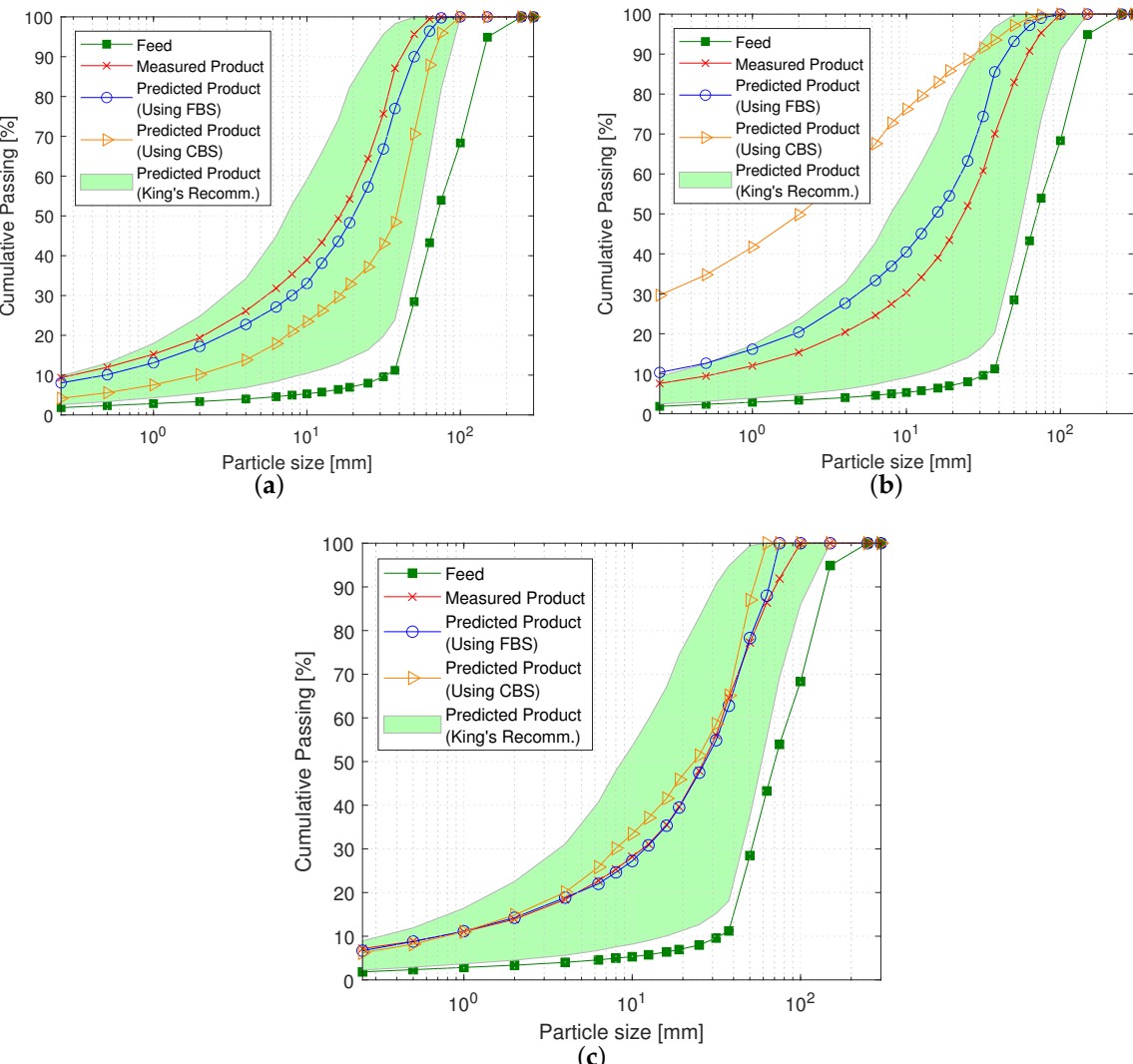

**Figure 7.** Measured and predicted PSD curves in Scenario 2 for the Serra Leste datasets. (**a**) PSD predicted using the model calibrated on datasets 2 and 3. Validation with Dataset 1. (**b**) PSD predicted using the model calibrated on datasets 1 and 3. Validation with Dataset 2. (**c**) PSD predicted using the model calibrated on datasets 1 and 2. Validation with Dataset 3.

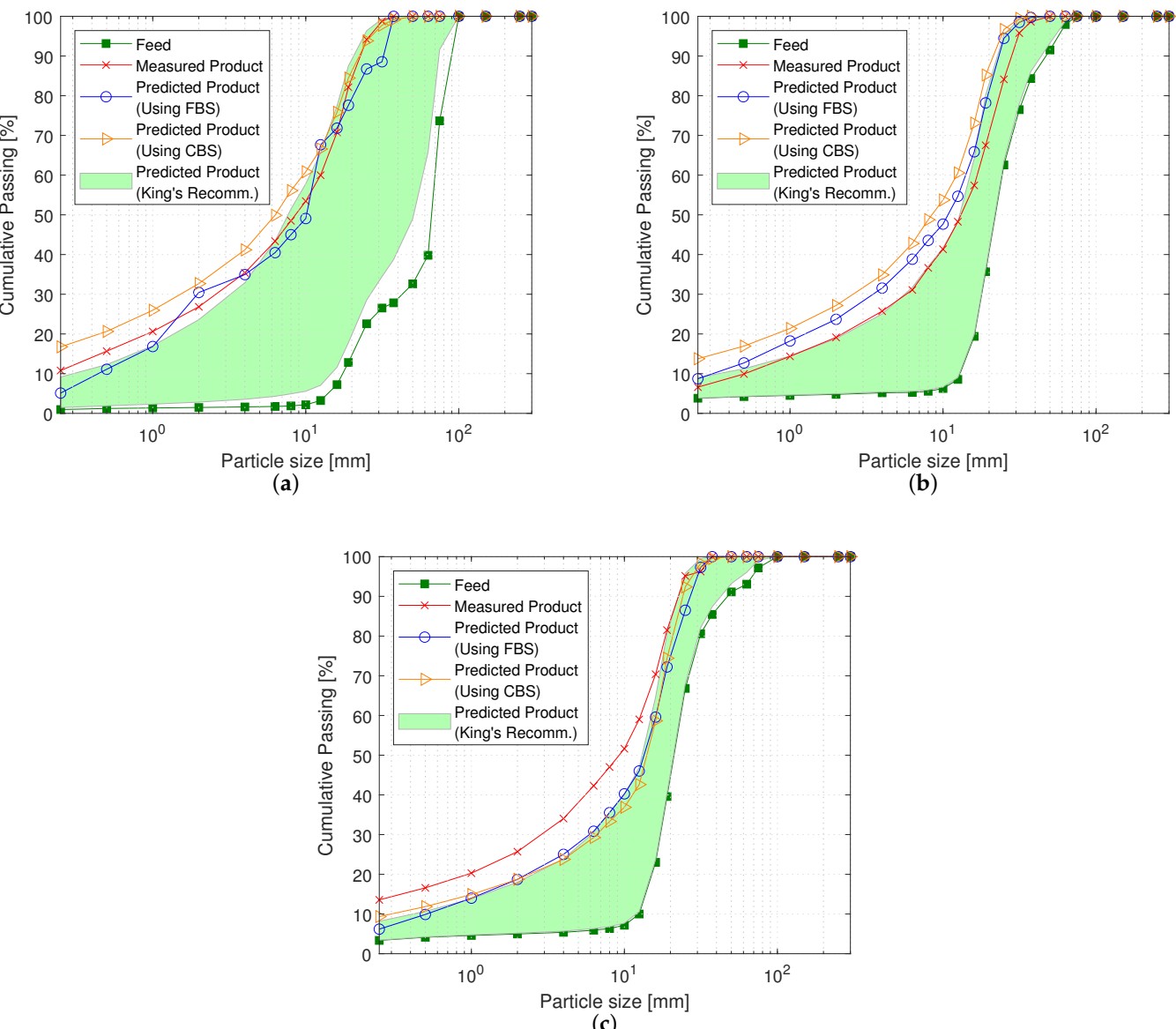

**Figure 8.** Measured and predicted PSD curves in Scenario 2 for the S11D datasets. (**a**) PSD predicted using the model calibrated on datasets 4 and 7. Validation with Dataset 5. (**b**) PSD predicted using the model calibrated on datasets 4 and 5. Validation with Dataset 6. (**c**) PSD predicted using the model calibrated on datasets 5 and 6. Validation with Dataset 7

Using the FBS, the SSE values obtained for the product PSD prediction in the validation step for datasets 1, 2, and 3 (Serra Leste) are 487.16, 1488.50, and 76.37, respectively. For datasets 5, 6, and 7 (S11D) the SSE values obtained for the product PSD prediction in the validation step are 393.64, 573.11, and 1151.30, respectively. Note that although the SSE values obtained in this scenario are relatively high, they nevertheless would not be a problem for certain practical applications, such as control applications.

Using the CBS, the resulting calibrated parameters are those showed in Table 7. The SSE values obtained in the validation step for datasets 1, 2 and 3 (Serra Leste) are 6249.20, 2007.430, and 541.09, respectively. The SSE values obtained in the validation step for datasets 5, 6, and 7 (S11D) are 433.67, 1658.00, and 1279.7, respectively.

**Table 6.** Calibrated parameters in Scenario 2.

| Datasets | $\alpha_0$ | $\alpha_1$ | $\alpha_2$ | $\alpha_3$ | $\beta_0$ | $\beta_1$ | $\beta_2$ | $\beta_3$ | $\gamma_0$ |
|---|---|---|---|---|---|---|---|---|---|
| 2 and 3 | 0.431 | 0.249 | 0 | 0.096 | 0 | 2.507 | 0 | 0 | 1.695 |
| 1 and 3 | 0 | 7.481 | 0.277 | 0 | 0 | 2.451 | 0 | 0 | 2.636 |
| 1 and 2 | 0 | 0.950 | 0 | 0 | 0 | 0 | 0.082 | 0 | 3.000 |
| 4 and 7 | 0.021 | 0.391 | 0 | 0.109 | 0.017 | 1.309 | 0.030 | 0 | 3.000 |
| 4 and 5 | 0.193 | 0.486 | 0 | 0.005 | 0.033 | 1.703 | 0 | 0 | 3.000 |
| 5 and 6 | 0.387 | 0.454 | 0 | 0.029 | 0.206 | 1.693 | 0 | 0 | 3.000 |

**Table 7.** Calibrated parameters in Scenario 2 (CBS).

| Datasets | $\alpha_0$ | $\alpha_1$ | $\alpha_2$ | $\alpha_3$ | $\beta_0$ | $\beta_1$ | $\beta_2$ | $\beta_3$ | $\gamma_0$ | $\Phi$ | $\delta$ | $\sigma$ |
|---|---|---|---|---|---|---|---|---|---|---|---|---|
| 2 and 3 | 0.024 | 2.127 | 0.303 | 2.380 | 0 | 3.500 | 0 | 0 | 2.275 | 1.000 | 0.534 | 75.929 |
| 1 and 3 | 0.022 | 29.596 | 1.410 | 2.276 | 0 | 3.218 | 0 | 0 | 3.000 | 1.000 | 0.514 | 237.607 |
| 1 and 2 | 0 | 0.500 | 0 | 0 | 0 | 0 | 0.067 | 0 | 2.688 | 0.016 | 0.487 | 242.818 |
| 4 and 7 | 0 | 0.642 | 0.011 | 0 | 0.047 | 0.096 | 0.093 | 0.379 | 3.000 | 1.000 | 0.343 | 577.232 |
| 4 and 5 | 0 | 0.858 | 0.051 | 0.192 | 0 | 0.397 | 0.094 | 0.076 | 3.000 | 1.000 | 0.408 | 764.263 |
| 5 and 6 | 0 | 0.903 | 0.035 | 0.053 | 0.001 | 0.813 | 0.109 | 0. | 3.000 | 1.000 | 0.417 | 644.004 |

### 5.2.3. Scenario 3

The goal of this scenario is to investigate the prediction capabilities of the Serra Leste cone crusher model calibrated in Scenario 1 for different CSS values. Specifically, product PSD values were calculated for CSS values of 37, 39, and 40 mm, which are in between the CSS values of the datasets used for calibration. Although no datasets obtained at the CSS values considered for prediction are available for model validation, it is expected that the relative ordering of the predicted product PSD curves should be consistent with that of the cone crusher CSS values. Figure 9 shows the measured PSDs for the CSSs used in the calibration step (35 mm, 38 mm, and 41 mm, marked in the legend with a '*' symbol) and the predicted PSDs for the intermediate CSSs calculated using the calibrated model from Scenario 1. The inset in the bottom-right part of the figure shows a detail view of the region around the size at which 80% of the product passes (*P*80), and Table 8 shows the respective *P*80 value for each CSS considered, where $\hat{P}$80 is the predicted value and *P*80 is the measured value. The fact that the $\hat{P}$80 values also appear in ascending order when the CSS values are arranged in ascending order confirms that the calibrated model can be useful for PSD prediction for intermediate CSS values.

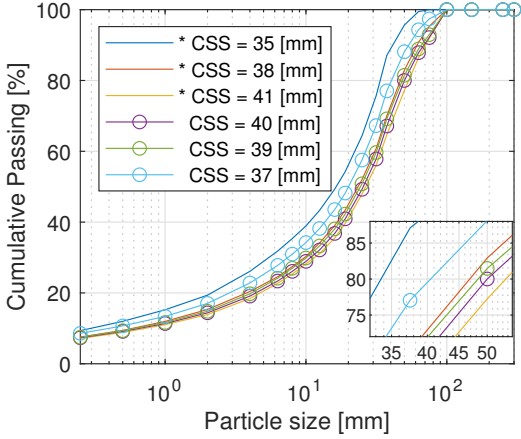

**Figure 9.** Measured PSDs for CSS = 35 mm, 38 mm and 41 mm. Predicted PSDs for CSS = 37 mm, 39 mm and 40 mm.

**Table 8.** *P*80 values in Scenario 3.

| CSS [mm] | *35 | 37 | *38 | 39 | 40 | *41 |
|---|---|---|---|---|---|---|
| $\hat{P}$80 [mm] | 34.66 | 40.84 | 46.95 | 48.51 | 50.00 | 52.39 |
| $P$80 [mm] | 33.78 | - | 47.16 | - | - | 53.90 |

The results presented in this scenario show how the calibrated cone crusher model can be used in dynamic simulations to predict product PSDs under operating conditions that do not match those considered in the calibration step. This characteristic is important because it enables the exploration of different operating conditions with the aim of optimizing certain performance criteria of the cone crusher, such as the throughput or product PSD.

In this scenario, only the sensitivity of the model to the CSS was studied. To obtain more comprehensive results, a similar study should be performed for every operating variable that affects the cone crusher model. Only in this way will it be possible to fully evaluate the robustness of the calibrated model. Nonetheless, the results presented here demonstrate that the model successfully predicts the expected trend of *P*80 with respect to variations in the CSS of the cone crusher, with an average prediction error of 1.97%, as shown in Table 8.

## 6. Conclusions

The calibration of a cone crusher model using industrial data was studied in this paper. In the model, the PSD of the product stream is calculated based on the PSD of the feed and the classification and breakage functions. Initially, the classification and breakage functions were defined based on widely used and accepted models from the literature. In our proposed strategy, the classification function, which has nine calibration parameters, remains unchanged. The breakage function, on the other hand, is altered to increase the number of decision variables in the calibration problem. Specifically, all elements of the breakage matrix are considered as parameters, whereas the calibration strategy applied in the literature relies on a condensed version of the breakage matrix in which the elements are expressed as a function of only three parameters. This modification increases the number of calibration parameters in the breakage function from three to $n_f(n_f+1)/2$, where $n_f$ is the number of classes used to characterize the material stream inside the cone crusher. The calibration problem is cast as an optimization problem based on the least squares method. The cost function is defined as the SSE between the predicted and measured product PSD values, and upper and lower bounds on the calibration parameters are enforced. Here, the resulting optimization problem was solved using an SQP algorithm. Notably, increasing the number of decision variables impacts the solution time for the optimization problem: we observed a difference of at most 179 s. Three calibration scenarios were analyzed, in which different combinations of calibration and validation tests were presented. The results show that the proposed calibration strategy is capable of dealing with sensitivity to the CSS. This finding indicates that increasing the number of degrees of freedom in the breakage function leads to a greater impact on the effect of the CSS in the crusher model. Due to the unavailability of additional data collected under different operating settings, it was not possible to investigate the effects of the liner properties, eccentric throw, or throughput on the calibration performance. Nonetheless, the proposed calibration method was found to enable the prediction of product PSDs for different CSS values in accordance with the expected results. One of the main advantages of the proposed methodology is that the calibration process is based only on data that are already available from the plant, namely, flow rates, throughput, CSS and PSD values. Additional laboratory tests are not required. The resulting cone crusher model is a strong candidate for use in dynamic simulations of crushing plants for process control applications, flowsheet design, and operator training. Future work will include model calibration and validation against more diverse datasets, permitting the exploration of the effects of other operating variables, such as liner properties and eccentric throw, on the model.

**Author Contributions:** Conceptualization, R.A.D., T.A.M.E., A.S.Y., M.T.d.S., L.P.C.; methodology, R.A.D., T.A.M.E., A.S.Y., M.T.d.S., L.P.C.; software, R.A.D.; validation, R.A.D.; formal analysis, R.A.D., L.P.C.; investigation, R.A.D.; resources, T.A.M.E.; data curation, R.A.D. and M.T.d.S.; writing—original draft preparation, R.A.D., A.S.Y., M.T.d.S., L.P.C., T.A.M.E.; writing—review and editing, A.S.Y., R.A.D., M.T.d.S., T.A.M.E.; visualization, R.A.D.; supervision, T.A.M.E., L.P.C.; project administration, T.A.M.E., L.P.C.; funding acquisition, T.A.M.E., L.P.C. All authors have read and agreed to the published version of the manuscript.

**Funding:** This research was funded by the Brazilian agencies CAPES (Finance Code 001) and CNPq (grants 402759/2018-4 and 444425/2018-7).

**Acknowledgments:** The authors are grateful for the support provided by the Vale S.A, the Instituto Tecnológico Vale and we would like to also thank Khalid Atta for his advice on dynamic modeling of cone crushers.

**Conflicts of Interest:** The authors declare no conflicts of interest.

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
