# Peer review of "Calibration and Validation of a Cone Crusher Model with Industrial Data"

_minerals, doi:10.3390/min11111256_

Round 1
Reviewer 1 Report
The paper can be published
Reviewer 2 Report
The reviewer would like to thank the authors for a detailed response. This is a very good paper! Below are a few issues to consider to the satisfaction of the editor.
The reviewer is satisfied with the response of the authors. However, it is important that each of the concerns are addressed directly in the article. Any items where I as reviewer am confused will be a potential point of confusion for other readers.
Therefore, please indicate where in the paper the following comments were addressed:
Response to Author Replies for:
- Comment 4: State explicitly that both CBS and FBS make use of the optimisation routine.
- Comment 5: Please state explicitly in the paper that F80 is not considered in the modelling exercise. Any influences from F80 is a result of the model itself.
- Comment 6: Remember that you are using a very large range of parameters. You are using far more fitting parameters than data points. Comment explicitly in the paper on the variation in the parameters.
- Comment 7: (On a personal note, the reviewer is still not convinced by that the model is validated in terms of its dynamic nature. Until the authors show a time-response plot, little can be said about the dynamic predictive abilities of the model. However, this is probably something to address in a future paper and does not need to be commented on here.)
- Comment 9: Eq. (3)
The units of the left hand side of equation is ton/h. Therefore, the right hand side should also be ton/h. However, if zeta and the elements S, B and C are dimensionless, then the final term in (3) has units ton. The dimensions for these variables are not correct.
Author Response
Please see the attachment.

This manuscript is a resubmission of an earlier submission. The following is a list of the peer review reports and author responses from that submission.
Round 1
Reviewer 1 Report
The current paper presents a new approach of calibrating cone crusher model with industrial data from iron mines. In overall, the quality of the paper is sufficient and it can be published after the authors will address a minor comments as follows:
- I would suggest the authors add additional keyword: "cone crusher".
- page 1, line 17: It is not clear what is the concept of "Industry 4.0". Please add a short description.
- Section 3: I would suggest the authors add a schematic view of a typical cone crusher and describe the principle of its operation.
Reviewer 2 Report
The submission presents an optimisation strategy to obtain the model parameters for a cone crusher model from industrial data. The authors expand the optimisation decision vector to include the breakage rates. The strategy is used in two cases studies where the model is validated with industrial data.
The article is very-well written and a pleasure to read. The results are interesting and the article has the potential to be a worthy contribution to the literature. However, there are a few areas of concern related to the results analysis and the conclusions.
The reviewer recommends major revision before publication. The aspect to be revised by the authors are as follows:
1. Validation scenarios
The reviewer is concerned that the validation exercise does not provide sufficient evidence of the generalisation abilities of the model. If the aim of the model is for control purposes, then a plant would like to use the model to predict circuit operation at different operating conditions. Thus, a good model may be fitted to one operating condition and is then capable to generalise accurately to other operating conditions.
The reviewer is not sure why Scenario 1 and 2 is included. If you "validate" a model against the same data used to fit the model, you are simply showing the accuracy of the fit. The question is rather to what extent you can fit the model to for example dataset 1, and accurately predict the data in datasets 2 and 3?
In contrast, Scenario 3 is a good validation exercises as the same data is not used for both fitting and validation. However, the SSE for the prediction is fairly high. To show that the model developed via the FBS method is an improvement on the CBS method, one would need to compare the SSEs for these two methods.
Note, the authors did not present a new model. Rather, they present a new approach to obtain model parameters. It is therefore important to show that this new approach (FBS) gives better results than the old approach (CBS). Although the FBS may fit the model better than CBS as shown in Table 2, it does not mean the FBS method will generalize over a wide range of operating conditions better than the CBS method. The reviewer is of the opinion that the validation exercises should also include the CBS optimisation results.
(The authors show in green in their figures the range of possible results of the CBS parameters. In many cases this overlaps with the FBS prediction. This indicates that the CBS parameters can give a similar prediction. The CBS parameters must be found using the minimisation routine and not via trail and error as mentioned in 387. It would be an unfair to compare trail and error results to the optimisation routine.)
2. Prediction capabilities (Scenario 4)
The authors claim in lines 436 that "the model can be used in dynamic simulations to predict product PSD". However, the authors only evaluated CSS. The model includes TPH and F80 as well. Therefore, the model can only trusted to show for variations in CSS.
The model cannot be regarded as accurate for TPH and F80 unless there is specific validation data to support this. In this regard the S11D data can only be used to validate the model for changes in F80 as this is the only variable that changes for data sets 4-7.
Since F80 is the only variable that changes in data sets 4-7 as shown in Table 1, it is curious to see such a large variation in calibrated parameters in Table 5 for parameters not associated with F80.
3. Dynamic vs steady-state simulation
The authors make the following statements:
- Line 10: "candidate for use in various dynamic simulation applications"
- Line 436: "can be used in dynamic simulations to predict product PSDs"
The model presented in (4) and (5) is not a dynamic model. It is a steady-state model. The authors need to clarify what they mean by "dynamic simulation" as the model used in the article is not dynamic.
Although (3) is a dynamic model, this model is not used. Also, it is not clear how (4) is obtained from the steady-state solution to (3) (see line 194-195). The ore hardness parameter and S in (3) is discarded in (4) without explanation. (Reference [5] develops (4) from a block diagram similar to Fig. 1, and does not show how (4) is obtained from (3). Similarly, [15] claims that (4) can be obtained from (3), but does not explain what happens to S or the ore hardness parameter.)
4. Laboratory analysis vs on-line data
The authors conclude in lines 477-479 that:
"One of the main advantages of the proposed methodology is that the calibration process is based only on data that are already available from the plant, namely, flow rates, throughput, CSS and PSD values. Additional laboratory tests are not required."
However, in lines 178-179 the authors state that:
"This value is determined by the number of sieves used in laboratory tests to define the PSD of the stream."
The calibration method requires the PSD of the product stream. It is not clear if this data is measured by means of an online analyser or if it is obtained from a sampling campaign.
If the data is available from an online analyser, why did the authors not consider the dynamics of the PSD as given by the (3)? If the data is not measured online, how is the PSD measured at a plant without a laborious laboratory sampling campaign?
Other Major and Minor Comments:
192-197: What are the units for the variables in eq. (3) and (4)?
Eq. (7) and (8): Please explain in brief why alpha_2 and Beta_4 have negative signs in front. Is this a heuristic modelling assumption in [36]?
215-223: Preferably mention (10) to (12) in that order in the text, or reshuffle the order of the equations. It is awkward to mention (12) before (10) and only (11) much much later.
229: "Equations (10) to (12) express...":
238: "we propose to calibrate every element of the lower triangular breakage matrix." (It should be clear that only the lower triangular elements are calibrated and not every entry in the matrix.)
240: The brackets for (11) should not go over two lines.
246: "based on the lower triangular breakage matrix, despite lacking..."
262-274: n_d (The variable nd should have the d as a subscript.)
Eq. (14): Preferably typeset "\all i \in CL" closer to the left and not so far to the right.
283: Parameters alpha_4, beta_4 and beta_5 are removed from theta_1. Although the authors mention that these parameters are not used later on, these parameters should be included in the search vector. Otherwise it is somewhat confusing at first reading. A similar comment applies to theta_2 in line 293.
285: Please give the equation number used to model \hat{P}.
289-290: "established in the literature." Please provide a reference to the exact "literature" referred to. Otherwise, please refer back to the literature study section earlier in the paper.
290-291: "in (12) are calibrated."
In earlier paragraphs equations are referenced only by their number in brackets (see 285-288). However, from this line forward all equations are introduced as "equation (x)". Please be consistent. (See also 307, 325, 326 etc.).
303-304: "the calibration of the cone"
Eq. (18): The decision variables for FBS and CBS are not the same. Therefore, the reviewer is of the opinion it may be best to split the lower and upper bounds equations in (18) such that the FBS variables and CBS variables are not mixed in one vector.
371: "indicates that the product PSD"
Should this not be the "feed PSD"? The parameter alpha_3 is associated with F80 in (7).
Table 3: What is interesting for Scenario 1 is not the ability to fit, but the range of the parameter values given in Table 2. There is a large variance in Beta_3 even though F80 remains constant in datasets 1-3. Can the authors comment on this? (Why are the parameter values for datasets 4-7 not shown in Table 2?)
392-394: The authors state that because alpha_3 is nonzero, the model accounts for changes in F80. However, beta_3 is zero and is also associated with F80? Similarly, alpha_2 is 0 in Table 4 in Scenario 2 even though TPH varies for datasets 1-3.
(One cannot assume alpha_2 in Table 3 is negligible because it is smaller than the other alphas. To make such a conclusion one would first need to scale the variables in (7)-(8) so that the range of sizes of each alpha and beta are equal.)
Table 4: Just to make the article more clear, maybe label "Serra Leste" as "Serra Leste (1 to 3)". This makes the associated dataset clearer. Similarly, maybe label "S11D" as "S11D (4 to 7)".
Table 5: Comment on the range of the values obtained between the different data sets. For example, why is there a spread of lambda_0 values for 1-3, but lambda_0 is constant for 4-7?
Page 21 and 22: Why are there empty pages?